



# Three principal components describe the spatiotemporal development of meso-scale ionospheric equivalent currents around substorm onsets

Liisa Juusola[1], Ari Viljanen[1], Noora Partamies[2], Heikki Vanhamäki[3], Mirjam Kellinsalmi[1], and Simon Walker[4]

[1]Finnish Meteorological Institute, Helsinki, Finland
[2]Department of Arctic Geophysics, University Centre in Svalbard (UNIS), Longyearbyen, Norway
[3]University of Oulu, Oulu, Finland
[4]University of Bergen, Bergen, Norway
**Correspondence:** Liisa Juusola (liisa.juusola@fmi.fi)

**Abstract.** Substorms are a commonly occurring but insufficiently understood form of dynamics in the coupled magnetosphere-ionosphere system, associated with space weather disturbances and auroras. We have used Principal Component Analysis (PCA) to characterize the spatiotemporal development of ionospheric equivalent currents as observed by the International Monitor for Auroral Geomagnetic Effects (IMAGE) magnetometers during 28 substorm onsets identified by Frey et al. (2004).

Auroral observations were provided by all-sky cameras. We found that the equivalent currents can typically be described by three components: a channel of poleward equivalent current (wedgelet), a westward electrojet (WEJ) associated with an auroral arc, and a vortex. The WEJ and vortex are located at the equatorward end of the channel, which has been associated with Bursty Bulk Flows (BBFs) by previous studies. Depending on its polarity, the vortex either indents the WEJ and arc equatorward, or bulges the WEJ poleward while winding the arc into an auroral spiral. In addition, there may be a background

current system associated with the large-scale convection. The dynamics of the WEJ, vortex, and channel can describe up to 95% of the variance of the time derivative of the equivalent currents during the examined 20 min interval. Rapid geomagnetic variations at the substorm onset location, which can drive Geomagnetically Induced Currents (GIC) in technological conductor networks, are mainly associated with the oscillations of the WEJ, which may be driven by oscillations of the transition region between dipolar and tail-like field lines in the magnetotail due to the BBF impact. The results contribute to the understanding

of substorm physics and to the understanding of processes that drive intense GIC.

## 1 Introduction

An auroral substorm typically starts with a brightening of an auroral arc followed by poleward expansion of the aurora (Akasofu, 1964). The leading edge of this poleward and westward expanding bulge is called the Westward Traveling Surge (WTS) (Akasofu et al., 1965b, 1966). The WTS is considered as the visual manifestation of the upward field-aligned current (FAC)

of the of the Substorm Current Wedge (SCW) (McPherron et al., 1973). A number of explanations have been proposed to



describe the sequence of events that leads to the substorm onset and the formation of the WTS and SCW. A thorough review of substorm physics is provided for example by Kepko et al. (2015).

When auroral data are not available, ground magnetometers in the nightside auroral region are often used to determine the substorm onset time and location (e.g., Newell and Gjerloev, 2011). The onset is then identified as a sudden decrease in the
horizontal magnetic field component ("negative bay"), which is caused by an intensifying Westward Electrojet (WEJ) (Akasofu et al., 1965a). Historically, the WEJ, associated with the Cowling channel (Coroniti and Kennel, 1972), has been assumed to close the idealized upward and downward FAC of the SCW at the edges of the dipolarized region in the magnetotail (McPherron et al., 1973). Although spacecraft have observed large-scale regions of net upward and downward FAC consistent with the SCW, these regions have also been observed to have significant substructure, consisting mainly of north-south aligned, oppositely
directed FAC (Forsyth et al., 2014). The north-south orientation of the FAC sheets is in contrast to the east-west orientation of the pre-onset aurora, but agrees with the observation of north-south streamers in the expansion phase bulge (Partamies et al., 2006). Furthermore, the upward FAC associated with the WTS has been shown to close mainly locally, with only about a third of the FAC diverted via the Cowling channel to remote current closure farther east (Amm and Fujii, 2008). It has been suggested that the SCW may consist of several wedgelets associated with bursty bulk flows (BBFs) in the magnetotail (e.g.,
Lyons et al., 2012). About 1/3 of substorm onsets have been shown to occur near the nightside auroral region called the Harang Discontinuity (HD), where the Eastward Electrojet (EEJ) changes into the WEJ (Weygand et al., 2008).

The upcoming Electrojet Zeeman Imaging Explorer (EZIE) mission (e.g., Laundal et al., 2021; Yee et al., 2021b, a) will investigate the small-scale structure and evolution of the auroral electrojet segment of the SCW and its possible modulation by wedgelets. In the meantime, the ground-based International Monitor for Auroral Geomagnetic Effects (IMAGE) magnetometer
network can shed light on the spatiotemporal development of meso-scale (100–1000 km) currents associated with the substorm auroral electrojet. This topic is also of interest from the point of view of geomagnetically induced currents (GIC) that are induced in technological conductor networks by rapid geomagnetic variations. Such variations occur often during substorm onsets when the amplitude of the WEJ increases rapidly (Viljanen et al., 2006). Recently, e.g., Juusola et al. (2023) and Milan et al. (2023) have deepened the analysis of major GIC events, but it is still partly unclear which processes in the ionosphere
and magnetosphere drive the most intense GIC.

Two components with different temporal characteristics have been identified in the expansion of the substorm auroras (Akasofu, 1964; Gjerloev et al., 2008; Lyons et al., 2013), indicating quasi-independent underlying processes. Initial auroral brightening rapidly expands east and west from the onset arc. The poleward and westward expansion of the bulge and WTS, on the other hand, initiate later, and the westward motion of the WTS is slower than the azimuthal brightening spreading from the
onset arc. A relatively weak depression of the horizontal ground magnetic field component is associated with the onset arc brightening and a delayed and substantially larger depression with the WTS (Lyons et al., 2013). Midlatitude Pi2 pulsations correspond to the poleward expansion (Ieda et al., 2018). It has been suggested that the auroral zone negative bay may not correctly time the onset of the substorm expansion phase (e.g., Lyons et al., 2012; Ieda et al., 2018). Another practical challenge associated with identification of substorm onset, especially from regional magnetic field observations, is to distinguish
between local substorm onsets and substorm activity expanding to the observed region from a remote onset region. The rapidly



varying ionospheric currents around substorm onsets induce strong telluric currents in the conducting ground (Tanskanen et al., 2001; Juusola et al., 2020), which further complicate the identification. Careful examination of the ionospheric currents as seen from the ground around substorm onsets identified from auroral observations should help to devise a way to distinguish between these two cases and help to determine whether the timing of the onset can be reliably determined from magnetic field observations.

We will use IMAGE magnetometer observations around local substorm onsets identified from global auroral images in 2000–2002 to study the spatiotemporal development of meso-scale ionospheric equivalent currents during these events. Our aim is to identify typical equivalent current behavior and associated ground magnetic field variations that occur during local substorm onsets. The results can contribute to the understanding of substorm physics and to the understanding of processes that drive intense GIC. They can be used to identify local substorm onset times and location from regional magnetic observations and to distinguish between local and remote substorm onset observations. The structure of the study is as follows: the data and methods are presented in Section 2, the results are presented in Section 3 and discussed in Section 4. The conlusions are summarised in Section 5.

## 2 Data and methods

### 2.1 Substorm onset list

We have used the substorm onset list compiled by Frey et al. (2004) based on global scale observations of the auroras at about 2 min cadence by the Far Ultra-Violet imager (FUV) on the Imager for Magnetopause-to-Aurora Global Exploration (IMAGE) spacecraft (the satellite and magnetometer network both have the same acronym). They used the following criteria to identify substorms between 19 May 2000 and 31 December 2002: (1) a clear local brightening of the aurora had to occur, (2) the aurora had to expand to the poleward boundary of the auroral oval and spread azimuthally in local time for at least 20 min, and (3) at least 30 min had to have had passed after the previous onset. The geographic and geomagnetic location of the substorm onset in the image of the initial auroral brightening was determined by finding the brightest pixel close to a visually determined center of the substorm aurora. Ieda et al. (2018) have suggested that substorm onsets identified from global auroral images do not necessarily correspond to the initial auroral brightening (Akasofu, 1964; Gjerloev et al., 2008; Lyons et al., 2013) but the subsequent poleward expansion.

Our criterium for selecting the substorm onsets that were within the most densely covered region of the IMAGE magnetometers was the following: the onset had to occur within the geographic latitude and longitude rectangle as defined by those of the northern Fennoscandia stations (SOR, TRO, AND, ABK, KIR, KIL, MUO, KEV, PEL, SOD, IVA, and MAS) (Figure 1) that were providing data at the time of the onset. The station located closest to the substorm onset site was selected as the onset station. There were 28 substorm onsets on the Frey et al. (2004) list that passed this selection. Considering the high magnetic latitudes of the analysis area, around $65^o$, the selected events are typically relatively quiet or moderate in terms of geomagnetic activity.



## 2.2 All-sky camera data

We have used auroral images at 557.7 nm wavelength (green emission) from the Magnetometers - Ionospheric Radars- All-Sky
Cameras Large Experiment (MIRACLE) All-Sky Cameras (ASCs) located at Kevo (KEV) and Kilpisjärvi (KIL) to observe
the local substorm onset auroras. The images are provided at a 20 s cadence, with an exposure time of 1 s. In order to compare
with ionospheric equivalent currents, the auroral intensity was projected to an optimal altitude near 110 km, which has been
determined by Whiter et al. (2013, 2023) using pairs of stations with overlapping fields of view. Only elevation angles above
70° are shown to cut out the parts with the largest positional uncertainty. Three out of the 28 events were found to have good
ASC data.

Upward FAC from the ionosphere to the magnetosphere is mainly assumed to be carried by the same electron population that
produces green auroras (e.g., Janhunen et al., 2000). Although green auroras only represent a portion of the energy spectrum
of the precipitating electrons, observation of such auroras can be used as an indicator of upward FAC.

## 2.3 Magnetometer data

We have used 10 s ground magnetic field measurements from the International Monitor for Auroral Geomagnetic Effects
(IMAGE) magnetometers. In 2000–2002, IMAGE consisted of 25–26 magnetometers in northern Europe (Fig. 1). The best
station coverage was between $64^o$ and $67^o$ magnetic latitude, where 12 stations (SOR, KEV, TRO, MAS, AND, KIL, IVA,
ABK, MUO, KIR, SOD, and PEL) were clustered within $7^o$ of magnetic longitude.

We have used the method by van de Kamp (2013) to subtract the long-term baseline (including instrument drifts, etc.), any
jumps in the data, and the diurnal variation from the variometer data. The remaining variation magnetic field consists of an
external part mainly due to ionospheric electric currents but with some magnetospheric contribution as well, and an internal
part due to induced telluric currents in the conducting ground. Both current systems are three-dimensional (3D), but according
to the equivalent current theorem, they can be replaced by divergence-free sheet currents on two spherical shells (e.g., Haines
and Torta, 1994), which produce the same magnetic field at the Earth's surface as the true 3D currents.

The equivalent current sheets could be placed at any location between the observation point and the entire 3D current
system. Because telluric currents can flow at any depth in the ground and the observations are made on the surface, there
are no alternatives, and the lower layer has to be placed just below the surface (cf. Juusola et al., 2020). These equivalent
currents cannot be interpreted in terms of true currents in the ground, but they can be used to remove the internal magnetic
field contribution from the observations. The upper layer could be placed at any height between the ground surface and about
90 km altitude, above which all ionospheric currents can be assumed to flow (e.g., Untiedt and Baumjohann, 1993). Placing
it at 90 km, however, has the significant advantage that the equivalent current density can be interpreted as the divergence-
free part of the true horizontal current density. There are two reasons for this: one is that the horizontal ionospheric currents
are concentrated in a relatively narrow layer close to 90 km altitude, and the other is that the magnetic field of the curl-free
horizontal current and radial field-aligned currents that flow between the horizontal current layer and the magnetosphere cancel
each other out below the horizontal current layer (Fukushima, 1976). Unlike originally suggested by Fukushima (1976), this



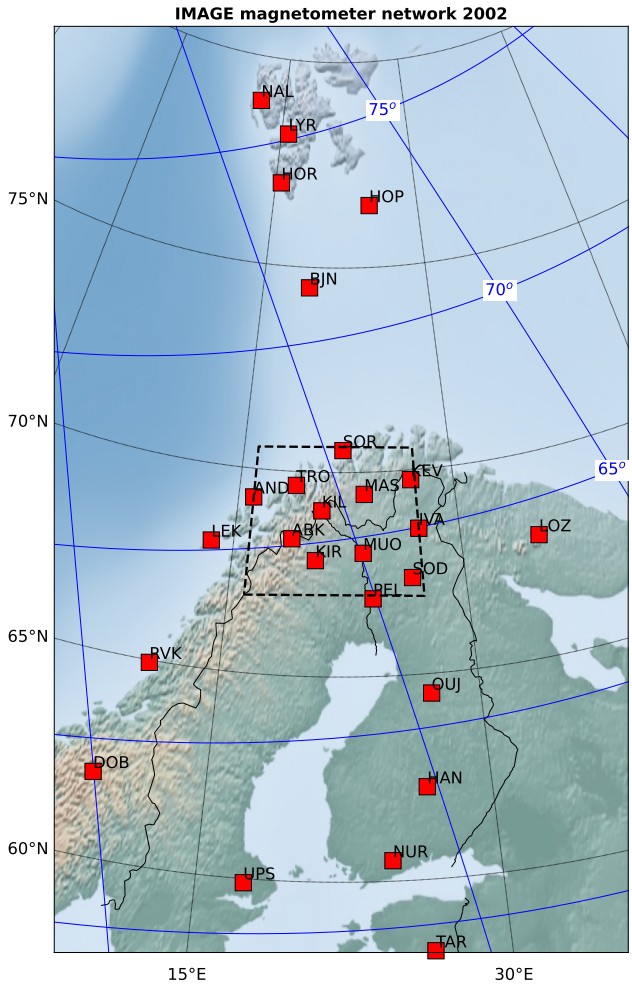

**Figure 1.** IMAGE magnetometer stations in 2002 used in the analysis and the area (dashed black rectangle) used for selecting substorm onsets. Quasi-dipole (QD) coordinates (Richmond, 1995; Emmert et al., 2010; Laundal et al., 2022) are indicated with the blue grid.

cancellation does not require uniform conductances in the ionosphere, but is valid for any conductance distribution (Amm, 1997). Thus, the equivalent current method can be used to separate the internal and external magnetic field contribution from ground magnetic field observations and to derive the divergence-free part of the horizontal ionospheric current density.

A key factor in the separation of internal and external magnetic field contributions is to use all three components of the
measured magnetic field. While either the horizontal components or the vertical component can be fully described by either the equivalent current sheet placed below or above the ground, reconstruction of all three component requires both equivalent current layers (Untiedt and Baumjohann, 1993; Vanhamäki and Juusola, 2018), and thus makes the separation possible. The separation of the external part eliminates the telluric contribution and makes physical interpretation in terms of primary ionospheric drivers easier (Juusola et al., 2020, 2023).





There are several methods that can be used to implement the equivalent current method. One option is the two-dimensional Spherical Elementary Current System (2D SECS) method (Amm, 1997; Amm and Viljanen, 1999; Pulkkinen et al., 2003a, b; Juusola et al., 2016; Vanhamäki and Juusola, 2020; Juusola et al., 2020), which we have used. In the 2D SECS method, the equivalent sheet current density is described in terms of divergence-free elementary current systems that consists of a localized curl at the pole and a globally distributed curl of the opposite polarity. In practise, the global parts typically cancel each other

out, limiting the curls to the region where the SECS poles are placed. Effects by curls outside the observed region are described by SECSs at the edges of the analysis grid, which thus needs to extend outside the grid used to visualize the results.

IMAGE data are provided in geographic coordinates but we have utilized the quasi-dipole (QD) coordinates (Richmond, 1995; Emmert et al., 2010; Laundal et al., 2022) to derive the ionospheric equivalent currents.

### 2.4 Interpreting divergence-free horizontal ionospheric currents

Ionospheric currents consist of horizontal currents in a narrow layer between about 100–150 km altitude (e.g., Untiedt and Baumjohann, 1993) and field-aligned currents that flow along the Earth's magnetic field lines between this altitude and the magnetosphere. In the thin-sheet approximation, the horizontal current density is estimated as a surface current $\boldsymbol{J}$ at about 100 km altitude. Ohm's law can then be written as:

$$\boldsymbol{J} = \underbrace{\Sigma_P \boldsymbol{E}}_{=\boldsymbol{J}_P} + \underbrace{\Sigma_H \hat{\boldsymbol{e}}_z \times \boldsymbol{E}}_{=\boldsymbol{J}_H} = \boldsymbol{J}_{CF} + \boldsymbol{J}_{DF}, \tag{1}$$

where $\boldsymbol{E}$ is the horizontal electric field, $\Sigma_H$ and $\Sigma_P$ are the Hall and Pedersen conductances, $\hat{\boldsymbol{e}}_z$ is a unit vector in the direction of the downward pointing radial geomagnetic magnetic field (approximately valid in the auroral region of the northern hemisphere), and $\boldsymbol{J}_H$ and $\boldsymbol{J}_P$ are the Hall and Pedersen current densities. Like any vector field, $\boldsymbol{J}$ can be expressed as a sum of a curl-free (CF) and a divergence-free (DF) part. Field-aligned current density is associated with the divergence of the horizontal current:

$$j_{||} = \nabla \cdot \boldsymbol{J} = \nabla \cdot \boldsymbol{J}_{CF}. \tag{2}$$

Assuming that the rotational inductive part of the electric field (e.g., Vanhamäki and Amm, 2011; Vanhamäki and Juusola, 2020) vanishes

$$[\nabla \times \boldsymbol{E}]_z = 0 \tag{3}$$

and that the conductances only have gradients parallel to the electric field

$$[\nabla \Sigma_P \times \boldsymbol{E}]_z = 0, \tag{4}$$

$$[\nabla \Sigma_H \times \boldsymbol{E}]_z = 0 \tag{5}$$





we can estimate that the divergence-free part of the horizontal current density equals the Hall current and that the curl-free part equals Pedersen current

$$\boldsymbol{J}_{DF} \approx \boldsymbol{J}_H = \Sigma_H \hat{\boldsymbol{e}}_z \times (-\boldsymbol{V} \times B_0 \hat{\boldsymbol{e}}_z) = -\Sigma_H B_0 \boldsymbol{V} \tag{6}$$

$$\boldsymbol{J}_{CF} \approx \boldsymbol{J}_P \tag{7}$$

where $\boldsymbol{V}$ is the convection velocity and $B_0$ is the strength of the Earth's magnetic field. With the additional assumption that the ratio between the Hall and Pedersen conductances is spatially constant

$$\alpha = \frac{\Sigma_H}{\Sigma_P} = const. \tag{8}$$

the curl of the divergence-free current density can be estimated to equal the field-aligned current density (positive down)

$$j_z = [\nabla \times \boldsymbol{J}_{DF}]_z \approx \alpha \cdot j_{||} \tag{9}$$

(Amm et al., 2002; Vanhamäki and Juusola, 2020). These assumptions are summarised in Table 1. Finally, a word of warning: although these approximations can help when interpreting ionospheric equivalent currents, it should be borne in mind that the assumptions behind them, especially Eq. 3, Eq. 4, Eq. 5, and Eq. 8, do not generally hold. Examples of such cases can be found in Untiedt and Baumjohann (1993). Furthermore, Figures 11–12 in Vanhamäki et al. (2009) show that the Hall current can have significant divergence (Eq. 6 does not hold), and Figure 8 in Vanhamäki et al. (2007) demonstrates that the rotational inductive part of the electric field can drive significant FAC during a substorm (Eq. 3 and, consequently, Eq. 9 does not hold).

## 2.5 Principal component analysis

We have used Principal Component Analysis (PCA) to find ionospheric equivalent current features that typically occur around substorm onsets. PCA is a method that can be used to represent a data matrix in terms of Empirical Orthogonal Functions (EOFs), or spatial modes, and their amplitudes, or temporal modes. PCA has been previously used, e.g., by Milan et al. (2015) to analyse FACs from the Active Magnetosphere and Planetary Electrodynamics Response Experiment (AMPERE) mesurements. We utilized Singular Value Decomposition (SVD) (Press et al., 1992) to perform PCA.

The ionospheric equivalent current density is often dominated by currents associated with the background convection pattern. Because we are interested in the changes that occur around substorm onsets, we do not apply the PCA directly to the ionospheric equivalent current density $\boldsymbol{J}$, but its time derivative. The time derivative is calculated as

$$d\boldsymbol{J}(t)/dt = [\boldsymbol{J}(t) - \boldsymbol{J}(t - T)]/T, \tag{10}$$

where $T = 10$ s is the time step of the data. Performing the PCA using $\boldsymbol{J}$ instead of $d\boldsymbol{J}/dt$ gives quite similar results, with the exception that the background current system is typically represented by the most significant EOF. The $d\boldsymbol{J}/dt$-based EOFs tend to be more distinct, probably, because they emphasize features with similar temporal characteristics.



**Table 1.** Summary of ground-based magnetic field interpretation.

| Action | Assumption | Validity |
| --- | --- | --- |
| Separating measured variation magnetic field into internal and external parts: $$B = B_{int} + B_{ext}$$ $$B_{int} \rightarrow J_{eq,int}$$ $$B_{ext} \rightarrow J_{eq,ext}$$ | Equivalent current theorem (Haines and Torta, 1994). | Valid. |
| Interpreting external equivalent current as divergence-free ionospheric current: $$J_{eq,ext} \approx J_{DF,ion}$$ | Radial field-aligned currents (Fukushima, 1976; Amm, 1997) and thin-sheet approximation (Untiedt and Baumjohann, 1993). | Approximately valid in the auroral region. |
| Estimating divergence-free ionospheric current as Hall current and curl-free ionospheric current as Pedersen current: $$J_{DF,ion} \approx J_H$$ $$J_{CF,ion} \approx J_P$$ | Vanishing rotational inductive part of the electric field and conductance gradients only parallel to the electric field (Amm et al., 2002). | Generally **NOT** valid (Untiedt and Baumjohann, 1993; Vanhamäki et al., 2007, 2009). |
| Estimating curl of $J_{DF}$ as field-aligned current: $$j_z = [\nabla \times J_{DF,ion}]_z \approx \alpha \cdot j_{||}$$ | In addition to the above: Spatially constant Hall-to-Pedersen conductance ratio $\alpha = \Sigma_H / \Sigma_P$. | Generally **NOT** valid. |

We note that Kruglyakov et al. (2022) represented ionospheric equivalent currents in the IMAGE region by the PCA to make their code feasible for fast computations. They needed 21 components to explain 99% of the variance of the magnetic field. As will be seen, our analysis requires a much smaller number, which is probably due to considering short events (20 min) of a particular type.

## 3 Results

In this section, we first present five example events, three with ASC data, and two without. The first three example events are selected according to the availability of the ASC data, and the last two cases based on the ambient background conditions during the events, such that all relevant types of substorm onsets are accounted for. Typical background conditions are either quiet or a WEJ or a HD, occasionally even an EEJ. At the end of the section we summarise the findings from all 28 events.



## 3.1 Example 1: Quiet background with ASC data

An overview of the time development of the ionospheric equivalent currents and ground magnetic field between $\pm 10$ min around a substorm onset on 18 December 2001 at 21:59:00 UT is provided in Figure 2. Fig. 2a shows the regional auroral electrojet indices derived from IMAGE data (Kauristie et al., 1996). The index derived from total (external + internal), external, and internal geomagnetic $B_x$ from all available IMAGE stations is drawn with black, blue, and red color, respectively. The thicker curves show the lower envelope (IL index) and the thinner curves the upper envelope (IU index). The vertical dashed line indicates the substorm onset time at 21:59:00 UT. While both the total and external IL show a small decrease at the substorm onset, the signature is quite weak. The internal IL has a larger amplitude than the external IL, indicating strong induction (Tanskanen et al., 2001). Before the onset, both IL and IU were almost zero, indicating quiet conditions with no significant background current. In the time period before the onset, the internal part is larger than the external part, but the amplitudes are so weak that this could be caused by inaccuracies in the baseline. Any offset or small error in the magnetic field of one station would not be consistent with the signatures of the nearby stations and would most likely be described as internal currents in the separation. Typically, the internal part is a few tens of % of the external part (Juusola et al., 2020). Because the indices only reveal currents in the east-west direction, the quiet conditions were confirmed by examining maps of the equivalent current density.

Fig. 2b–c show latitude profiles of external $j_z$ and its time derivative ($dj_z/dt$) as a function of UT along the longitude of the IMAGE station (AND) located closest to the substorm onset site. The purple curves in the plot indicate where $j_z$ changes its sign. Fig. 2b–c show an intensification of positive $j_z$ (red, proxy for downward FAC) poleward of the onset site and of negative $j_z$ (blue, proxy for upward FAC) equatorward of the onset site, as well as a small equatorward drift of the boundary between these two regions during the first minute after onset. During the next minute the region of upward $j_z$ intensifies and shifts poleward. This is followed by a more gradual shift back towards the equator.

Figure 3 shows maps of the ionospheric equivalent current density (arrows), its curl (color background), and ASC images at selected moments around the substorm onset. The left hand side column includes the ASC images, and the right hand side column shows the equivalent current density 10 s later, for comparison. Note that the scaling of the arrows and color background varies from panel to panel while the grey scale remains the same (black – 0 counts, white – 255 counts). The yellow star indicates the substorm onset location. The black vertical line passing through the nearest station (AND) indicates the locations from which latitude profiles are extracted to create the time series representations in Fig. 2.

At the time of the substorm onset (21:59:00 UT), there was an auroral arc at the onset location and some auroral structures poleward of the arc east of the onset location. The equivalent current pattern shows a WEJ with a region of upward $j_z$ at the location of the auroral arc. At 21:59:40 UT, the arc had developed a poleward bulge. A corresponding bulge had also appeared into $j_z$. At 22:40:00 UT, the auroral bulge had developed into a clear auroral spiral. The corresponding equivalent currents show an anticlockwise vortex with intense upward $j_z$ in the center. At 22:02:20 UT, the auroral spiral had mostly disappeared westward out of the field-of-view of the ASC. Only the tail of the spiral is still visible and corresponds to a region of upward $j_z$ in the equivalent currents. The eastward end of the spiral tail is first bent equatorward and then again poleward, following





**Figure 2.** (a): Regional auroral electrojet indices derived from IMAGE data (Kauristie et al., 1996) $\pm 10$ min around a substorm onset on 18 December 2001 at 21:59:00 UT. The index derived from total (external + internal), external, and internal geomagnetic $B_x$ from all available IMAGE stations is drawn with black, blue, and red color, respectively. The thicker curves show the lower envelope curve IL index and the thinner curves the upper envelope curve IU index. The vertical dashed line indicates the subtorm onset time. (b): Latitude profiles of external $j_z$ as a function of UT along the longitude of the IMAGE station (AND) located closest to the substorm onset site. The purple curves indicate where $j_z$ changes its sign. (c): Latitude profiles of external 10 s $dj_z/dt$.

230   the streamlines of the equivalent current vectors around a region of strong downward $j_z$. If $\boldsymbol{J}_{eq} = J_H$ (Eq. 6) is valid, the ionospheric plasma convection will be directed antiparallel to $\boldsymbol{J}_{eq}$.

Figure 4 shows the results of the PCA: the four most significant EOFs (EOF1–EOF4) of $d\mathbf{J}/dt$ and $dj_z/dt$ and their amplitudes. The time stamps refer to the onset time of the analysed event. The four EOFs together described 95% of the variance of the time derivatives during the analysed 20 min time interval. The interval length was selected so that it more or less included





the relevant dynamics after onset. The results of the PCA were not particularly sensitive to the length of the interval. This will be demonstrated later in section 4.5. Additionally, we show time integrals of the amplitudes in order to describe the development of the corresponding $J$ and $j_z$. EOF1 describes a WEJ, which is bent northward east and west of the onset location. EOF2 describes a north-south oriented channel of northward equivalent current, and EOF4 its westward motion and expansion. EOF3 describes a vortex of anticlockwise equivalent current with upward $j_z$ at its center. The vortex is located in the middle of the WEJ (EOF1) and in the middle of the southward end of the channel (EOF2, EOF4), where the northward equivalent current diverges eastward and westward. The amplitude of EOF1 shows decaying oscillations, which result in an intensifying WEJ at the time of the onset, followed by a period of EEJ, and later a persistent WEJ. The channel (EOF2) also shows oscillations, although less distinct than those for the WEJ, that result in a persistently positive integrated amplitude. The more extensive version of the channel (EOF4), follows with a small delay. The vortex in EOF3 intensifies strongly at the time of the onset and remains positive afterwards for the duration of the examined time interval.

Comparison with Fig. 3 shows that at the time of the onset (21:59:00 UT) the WEJ (EOF1) gave the strongest contribution to the equivalent current pattern. At 21:59:40 UT, the strongest contribution came from the vortex (EOF3) and the WEJ (EOF1), and at 22:00:40 UT, from the vortex (EOF3) and the EEJ (EOF1). At 22:02:20 UT, the channel (EOF2) had become the strongest contribution.

## 3.2 Example 2: Background Harang Discontinuity with ASC data

Figure 5 shows data for a substorm onset on 19 Feb 2002 at 20:13:00 UT in the same format as Fig. 2. The IL index (Fig. 5a) shows a depression around the onset, but the amplitudes remain relatively weak, only some tens of nT. The latitude profiles of $j_z$ and its time derivative at the onset location (Fig. 5b–c) show an intensification of upward $j_z$ at the time of the onset.

Although the IL and IU indices before onset were close to zero, examination of the equivalent current maps revealed a weak but persistent northward equivalent current typical for the Harang Discontnuity (HD) region. This background current system is still clearly visible at the time of the substorm onset at 20:13:00 UT in Fig. 6. In addition, a region of upward $j_z$ has started to intensify at the onset location, and an auroral arc has developed a poleward bulge.

At 20:15:00 UT, both the auroral bulge and anticlockwise equivalent current vortex have grown and intensified. At 20:18:00 UT, the center of the auroral bulge has moved westward and the structure has started to resemble a spiral. Similar to the previous example, the eastward tail of the spiral first curves equatorward and then tailward as it extends eastward, following the equivalent current pattern around a region of downward $j_z$. A similar pattern can still be seen at 20:22:20 UT, although the equivalent current north of the onset region has developed into a northwest-southeast oriented channel.

The PCA results are shown in Fig. 7. Similar to Example 1, the most significant EOFs include a vortex (EOF1), a WEJ (EOF2), and a channel (EOF3 and EOF4). At the time of the onset (cf. Fig. 6 at 20:13:00 UT), the WEJ (EOF2) gave the strongest contribution, although the anticlockwise spiral (clockwise in EOF1 but with a negative integrated amplitude) and channel (EOF3) were already intensifying. The WEJ (EOF2) remained the strongest contribution throughout the examined time interval. The vortex (EOF1), oscillated between negative and positive integrated amplitude, with the strongest negative





peak (i.e., anticlockwise vortex) at 20:15:00 UT and 20:18:00 UT. At 20:22:20 UT, there was a clockwise vortex and the contribution from the channel (EOF3) peaked.

### 3.3    Example 3: Quiet background with ASC data

The onset of our third example took place on 11 Mar 2002 at 21:06:00 UT and had similar background conditions as the first example, i.e., quiet (Figure 8a, confirmed by examining equivalent current maps). At the time of the onset, upward $j_z$ again intensified at the onset location (Fig. 8b–c).

The equivalent current pattern that developed at the onset (Fig. 9) was again a northwest-southeast oriented narrow channel of northwestward equivalent current. An auroral arc, located at the southward end of the channel, developed a northward bulge at the time of the onset (21:06:00 UT), which brightened and expanded during the next minutes. The auroral bulge coincided with intense upward $j_z$ west and south of the equivalent current channel. Unlike in the two previous examples, in this case the auroral bulge did not form a spiral structure. At 21:13:00 UT, an auroral structure that resembled the spiral tail in the previous examples still followed the equivalent current pattern, similar to the previous examples.

The four most significant EOFs from the PCA (Fig. 10) are somewhat more difficult to interpret in this case. It seems that EOF1 may describe both the vortex and WEJ and that EOF2 (negative integrated amplitude) describes the channel. EOF3 activates later and may represent westward motion of the channel and a WEJ that bulges equatorward. At the time of the onset (21:06:00 UT in Fig. 9), the strongest contribution came from EOF1 (vortex and WEJ). At 21:06:40 UT, EOF1 contribution peaked, and at 21:07:40 UT, EOF2 (channel) contribution. At 21:13:00 UT, EOF2 contribution had vanished, and there was a relatively strong contribution from EOF3 (channel and WEJ).

### 3.4    Example 4: Background EEJ

Figure 11 shows that the background current system for the substorm onset that took place on 24 Jul 2001 at 20:33:00 UT was an EEJ (positive IU in Fig. 11). This was confirmed by examination of the equivalent current maps. The IL index shows almost no indication of the substorm, but there is a weakening of the IU index around the substorm onset. The latitude profiles of $j_z$ and its time derivative (Fig. 11b–c) at the onset longitude show some weakening of the downward $j_z$ at the onset latitude, but this event does not resemble the previous examples.

Because there is no ASC data available for this event, Figure 12 only shows the equivalent current maps at selected moments of time around the substorm onset. At 20:31:00 UT, two minutes before the substorm onset, the dominant current system was the background EEJ. One minute later, at 20:32:00 UT, the equivalent current vectors around the substorm onset site begun to turn northward, and at the time of the substorm onset (20:33:00 UT), there was a clear channel of northward equivalent current at the onset location. The channel strengthened and persisted until 20:42:00 UT, after which it disappeared and the equivalent current pattern resumed the EEJ configuration.

The four most significant EOFs produced by the PCA in Figure 13 can again be interpreted in terms of the combined vortex and WEJ (EOF1 and EOF2), and channel (EOF3 and EOF4). Two minutes before the onset, at 20:31:00 UT, there was a small contribution from EOF2. At 20:32:00 UT, the contribution from EOF2 had increased, and in addition there was even stronger





contribution from EOF1. At the time of the onset (20:33:00 UT), EOF1 gave the strongest contribution to the equivalent current pattern. At 20:42:00 UT, contribution from the channel (EOF3) peaked.

### 3.5 Example 5: Background WEJ

The substorm on 27 Jul 2002 took place clearly later in MLT than the previous examples, around 03:30 MLT, and there was a
strong background WEJ. This can be seen in strong negative value of the IL index (Fig. 14a). The IL or IU indices do not show any clear substorm signatures, and the latitude profiles of $j_z$ and $dj_z/dt$ only reveal a weak poleward expansion of the boundary between upward and downward $j_z$. This was followed by two additional expansions at ~01:28:00 UT and ~01:31:00 UT.

Figure 15 shows the equivalent current pattern at four selected moments of time around the substorm onset: one minute before (01:22:00 UT), at the time of the onset (01:23:00 UT), two minutes later (01:25:00 UT), and at the end of the analysed
interval (01:27:00 UT). Unlike the previous examples, the equivalent current pattern shows very little changes due to substorm onset, probably, because the strong background WEJ masks them.

The PCA results, however, presented in Figure 16, again show similar patterns as those in the previous examples: a vortex (EOF1), a WEJ (EOF2), and a north-south oriented channel (EOF3 and EOF4). Before the substorm onset, at 01:21:00 UT and at 01:22:00 UT (cf. Fig. 15), the WEJ (EOF2) and channel (EOF3) gave the strongest contribution. At the time of the
substorm onset, at 01:23:00 UT, contribution from the vortex (EOF1) had started to increase, and it peaked at 01:25:00 UT, 01:28:00 UT, and 01:31:00 UT, corresponding to the little poleward expansions in Fig. 14b. At the end of the examined time interval, at 01:33:00 UT, the channel (EOF3) gave the strongest contribution.

### 3.6 Summary of all events

Table 2 summarises our visual examination of the PCA for all substorm onsets by Frey et al. (2004) that occurred in the region
most densely covered by the IMAGE magnetometers. The rows are ordered according to MLT, with premidnight events at the top and postmidnight events at the bottom. Because the results of this table are based on visual inspection of the PCA results, they are somewhat subjective and should be considered indicative.

The entries in the column labeled as "Background" confirm what the examples already showed: substorm onsets can occur in locally quiet conditions or in a background EEJ, Harang Discontinuity (HD), or WEJ equivalent current configuration, with
EEJ and Harang most likely in the pre-midnight sector and WEJ around midnight and in the post-midnight sector. Pure EEJ background conditions are quite rare: we only found one case, which was presented as Example 4. Weygand et al. (2008) have shown, using partly the same set of substorm onsets as we, that approximately 1/3 of auroral substorm onsets occur within or near the HD identified in the growth phase. They defined the HD as the transition from relatively strong eastward to relatively strong westward equivalent ionospheric currents. They also suspected that some of the events were not true substorm onsets
but pseudobreakups or poleward boundary intensifications.

The description of the equivalent current configurations, which the PCA results were interpreted to represent, indicate that the WEJ, channel, and vortex can be identified in the majority of the cases. In some cases (5 in Table 2) the interpretation was not clear, and these are marked with a question mark (?). In some cases the main part of the relevant equivalent current pattern





was clearly outside the magnetometer coverage (over the sea such that the details could not be resolved sufficiently), despite

the careful selection. In some cases, the interpretation was just not straightforward.

## 4 Discussion

### 4.1 Summary of observations

Figure 17a summarises our findings in the form of a simplified sketch. The equivalent currents around substorm onsets consist of three basic components: a channel of northward equivalent current (grey), a vortex of either anticlockwise or clockwise

equivalent current (blue), and a WEJ (black). The center of the vortex is located at the southward end of the channel, where the northward equivalent current diverges eastward and westward, and where the WEJ also flows. The region where the onset is typically located with respect to the equivalent current patterns is indicated with a red box.

Although the channel, vortex, and WEJ can be identified in the majority of the examined substorm onsets, they are not always perfectly separated into different EOFs. Especially the vortex and WEJ are often entangled, and as a result there is

either a poleward or equatorward bend in the WEJ. The poleward bend (e.g., EOF1 in Fig. 13) results when a WEJ is combined with an anticlockwise vortex, which strengthens the westward current poleward of the WEJ center and weakens it equatorward of the WEJ center, as illustrated in Fig. 17c. The equatorward bend (e.g., EOF1 in Fig. 4) results when a WEJ is combined with a clockwise vortex, which strengthens the westward current equatorward of the WEJ center and weakens it poleward of the WEJ center (Fig. 17d). The separate vortex in another EOF can then describe a change in the polarity of the vortex, for

example.

A typical time development is such that shortly before the substorm onset there is a WEJ combined with a clockwise vortex (proxy for downward FAC), decribing local equatorward drift of the WEJ. At substorm onset, the polarity of the vortex changes, and the anticlockwise vortex (proxy for upward FAC) describes the local poleward expansion of the WEJ. This is followed by decaying oscillations, where the polarity of the vortex alternates between negative and positive (e.g., EOF1 (vortex) in Fig. 7

and EOF1 (WEJ + vortex) in Fig. 4). The oscillating vortex then typically settles to positive values. The time development of the channel is typically such that it starts to intensify shortly before the substorm onset, and keeps intensifying until the end of the examined 10 min time interval.

Comparison of ionospheric equivalent currents and auroras in three examples indicates that before the substorm onset there is typically an auroral arc located in the middle of the WEJ with an equatorward bend (Fig. 17c). Ritter et al. (2004) have shown

that small-scale FACs, which might be associated with an auroral arc, are typically located in the middle of the larger-scale WEJ. At the substorm onset, the appearing anticlockwise vortex starts to bend the WEJ poleward and to wind the arc into a spiral (Fig. 17b) (Partamies et al., 2001). The spiral is thus associated with intense negative curl of the equivalent current density, which can be interpreted as upward FAC under the conditions outlined in section 2. As the channel equivalent current intensifies and the oscillations subside, the westward edge of the area of negative curl with which the spiral is associated expands westward

out of the field-of-view, taking the spiral with it. As the vortex changes polarity, producing again an equatorward bend in the WEJ, a matching bright bend can be observed in the auroras as well (e.g., Fig. 6 at 20:18:00 UT).



The observed WEJ, channel, and vortex and their typical time development are in good agreement with earlier observations, as outlined in the Introduction. Especially, the WEJ agrees with the east-west aligned pre-onset auroral arcs. The vortex associated with the auroral spiral agrees with the mainly local FAC closure associated with the WTS (Amm and Fujii, 2008). The channel agrees with the weaker remote current closure component of the WTS (Amm and Fujii, 2008) as well as the observation of north-south aligned FAC sheets within the bulge (Forsyth et al., 2014). The separation of the WEJ, vortex, and channel into different EOFs with different temporal behavior agrees with the observations of different temporal charateristics of oval and bulge auroras (Gjerloev et al., 2008; Lyons et al., 2013). Partamies et al. (2003) examined plasma flow and FAC distributions around a pseudobreakup spiral that took place during background WEJ conditions. The spiral was shown to be associated with a localized current wedge, tilted in the northeast-southwest direction, which is consistent with a tilted channel. Furthermore, the observed clockwise plasma flow around the spiral is consistent with a northward bulge in the WEJ.

## 4.2 A possible interpretation

Earthward BBFs in the plasma sheet of the magnetotail have been suggested to play a role in triggering substorm onsets (Nishimura et al., 2010). Moreover, an equivalent current channel similar to the one we have found as one of the three basic components of substorm onset equivalent current structures, has been suggested to be the ionospheric manifestation of fast earthward flows (Kauristie et al., 2003; Juusola et al., 2009). Such a signature is also called a wedgelet, because the FAC structure associated with it (downward FAC on the east flank of the channel and upward FAC on the west flank of the channel) resembles a smaller-scale version of the SCW. Thus, an interpretation, where BBFs play a significant role, could explain the observed equivalent current dynamics. The suggested interpretation only concerns the immediate spatial and temporal vicinity of the substorm onset, not the growth phase preceding it or the larger-scale dynamics that follow it.

In this scenario, a BBF from an X-line approaches the Earth before the substorm onset. As it reaches the transition region between the tail-like and dipolar magnetic field lines, it compresses the transition region, producing a localized indentation (1. in Fig. 17e). In the ionosphere, this would show as an intensifying, southward-drifting WEJ and an associated auroral arc immediately before the poleward expansion (Fig. 17d). Substorm onset occurs when the transition region cannot be compressed further in, and it bounces back (2. in Fig. 17e) (e.g., Chen and Wolf, 1999; Ohtani et al., 2009; Panov et al., 2010; Birn et al., 2011; McPherron et al., 2011; Juusola et al., 2013), corresponding to the poleward expansion in the ionosphere, as described by the anticlockwise vortex and auroral spiral (Fig. 17c). The oscillations of the transition region around its new balance point, between the magnetic and thermal pressure of the inner magnetosphere and the dynamic pressure of the flow, would produce the decaying oscillations of the vortex in the ionosphere, analogous to what happens on a global scale on the dayside when a solar wind pressure pulse hits the Earth's magnetopause (e.g., Juusola et al., 2010). As the fast flow intensifies, pushing the transition region further in (3. in Fig. 17e), and the X-line drifts tailward, the ionospheric channel expands and its equatorward end drifts equatoward (Fig. 17d). Abrupt intensification of the flow could also produce a new series of oscillations. The established flow would start to pile up magnetic flux against the transition region, which is sketched in Fig. 17e (3.) as dipolarized regions on both sides of the flow channel. The region on the duskside, which would correspond to negative curl in the ionosphere, might be associated with the westward propagation of the auroral spiral.



Out of the three basic current systems we have identified, the channel is the only one that indicates FAC closure in the east-west direction, in the manner of the SCW. The WEJ indicates current closure in the meridional plane, and the vortex symmetrically around the center. Technically, it is also possible that the field-aligned current suggested by the vortex could close farther away, outside the field-of-view of the magnetometers. The auroral spiral, which is associated with the vortex, may
be the beginning of the WTS. The appearance of the spiral seems to be associated with the appearance of the anticlockwise vortex at the equatorward end of the channel. Our analysis shows that the spiral starts to move west as the region of negative curl expands west together with the intensifying and expanding channel, but our analysis does not cover the westward and poleward propagation of the spiral.

## 4.3 Substorm onset as a driver of GICs

Substorm onsets have been shown to be among the most significant drivers of rapid geomagnetic variations (Viljanen et al., 2006), which in turn drive geomagnetically induced currents (GIC) in technological conductor networks (Viljanen et al., 2001). Hence, we consider the role of the three basic equivalent current components, the channel, vortex, and WEJ, would have in producing rapid geomagnetic variations at the substorm onset location. The sketch in Fig. 17a reveals that clearly the main contribution would be expected to be variations in the north magnetic field component due to the WEJ dynamics, depending
on exact substorm onset location.

As an example, we will examine the magnetic field components and their time derivatives at the onset location for the event that occurred on 18 Dec 2001 (section 3.1). Figure 18 shows the total, external, and internal geomagnetic north ($B_x$), east ($B_y$), and down ($B_z$) components and their 10 s time derivatives at station AND, which was located closest to the substorm onset site, $\pm 10$ min around the substorm onset. Although the magnetic field amplitudes are relatively weak, $dB_x/dt$ shows relatively
strong values exceeding $1 \, \mathrm{nTs}^{-1}$ in amplitude, which is often used as an indicative threshold for significant GIC (cf. Viljanen et al., 2006). Comparison with Fig. 4 reveals that external $B_x$ corresponds mainly to EOF1 (WEJ) and $B_z$ to EOF3 (vortex). The strongest peaks in $dB_x/dt$, $dB_y/dt$, and $dB_z/dt$, on the other hand, are mainly associated with EOF1, i.e., the intense, decaying oscillations of the WEJ. This is consistent with, e.g., Milan et al. (2023), who have shown that rapid magnetic field variations or "spikes" associated with substorm activity are mainly in the north-south direction. In agreement with Tanskanen
et al. (2001) and Juusola et al. (2020), Fig. 18 shows that the internal part of the magnetic field and its time derivative contribute significantly to the total magnetic field and its time derivative.

Considering the chaotic and apparently unpredictable nature of the time derivative of the ground magnetic field in general (Kellinsalmi et al., 2022), the percentage of the ionospheric $d\boldsymbol{J}_{eq}/dt$ variance that the four most significant EOFs can explain $\pm 10$ min around the substorm onset (Table 2) is surprisingly high. This means that the geomagnetic variations associated with
a substorm onset could maybe be forecasted based on the three basic equivalent current structures and their typical temporal behavior. The strongest time derivative signatures seem to be associated with the ionospheric manifestation of the oscillations of the magnetospheric transition region, possibly due to an impact of a BBF. This is in line with the result that the most rapid geomagnetic variations are associated with changes in the magnetospheric magnetic field configuration (Juusola et al., 2023). Combining this forecast of the ionospheric equivalent currents with fast 3D induction modeling (Kruglyakov et al., 2022)




would provide a tool for forecasting GIC due to a local substorm onset, considering the local ground conductivity distribution. However, predicting the exact substorm onset location would still remain a challenge, although it might be possible to estimate the strongest possible effect in an area of interest.

### 4.4  Substorm onset identification from regional magnetic field observations

While auroral images can be used to determine substorm onset time and location, using magnetic field observations is more
challenging. Magnetic field observations have the advantage of long, continuous time series, but while technically identification of the sharp drop in the north component, which has traditionally been used to identify substorm onsets, is relatively straightforward, it is not alone a reliable indicator of a local substorm onset. Sometimes such a change is nearly invisible as our Examples 4–5 illustrate. Substorm activity spreading to the observed region can also produce such a signature. It would be very useful to have a reliable means of identifying local substorm onsets from regional magnetic field observations. It now
seems that the PCA analysis could be used for that purpose, in two ways. The first possibility would be to perform the analysis for an interesting time interval and see if the channel, WEJ, and vortex make an appearance. The other possibility would be to decompose an extended time series of ionospheric equivalent current distributions in terms of the substorm EOFs and select as local onsets the periods when they can sufficiently well describe the distribution. However, this requires a dense magnetometer network capable of resolving the meso-scale equivalent current structures associated with the substorm onset. In case this is
not available, auroral observations would probably be needed for substorm onset identification.

### 4.5  PCA of a different event type and time interval

Considering the small area of our analysis region, it seems possible that an east-west current system, a north-south current system, and a vortex pattern could explain variations in the local equivalent current pattern more generally than around substorm onsets. Although this does not affect our conclusions, it might affect the usefulness of the PCA analysis in identifying local
substorm onsets. Examining this properly will require a separate study, but we will make a start by examining two examples. The first example is a longer time interval of the substorm onset of section 3.1. The overview Figure 19 is otherwise the same as Fig. 2, except for a longer time interval from $-30$ min to $+120$ min around the substorm onset. The longer time interval covers the entire substorm in the IMAGE local time sector, consisting of two activations: the first one around 22:00:00 UT and the second one after 22:30:00 UT. The PCA analysis (Figure 20) produces three EOFs (WEJ – EOF1, channel – EOF3
(signs of EOF and amplitude are swapped), vortex – EOF4), which are quite similar to the analysis of the 20 min time interval around the substorm onset. This is not particularly surprising, because the strongest time derivatives during a substorm onset would be expected to be associated with the onset. However, there is an additional EOF (EOF2), which describes a WEJ at the poleward edge of the bulge later during the substorm. As can be seen by comparing Fig. 19a and the amplitude of EOF2 in Fig. 20, sharp activations of this current system around 23:00:00 UT produce strong peaks in the internal and total IL index,
which are caused by induced currents in the sea surrounding Svalbard over which the westward current of EOF2 is located. These four EOFs can describe 92% of the variance of $d\boldsymbol{J}_{eq}/dt$ during the two hour time interval, which is only a little less than the 95% in section 3.1. Examining how much of the variance a certain number of EOFs can explain for time intervals of



varying lengths and for areas of different size, and how this information can be utilized in predicting $d\boldsymbol{B}/dt$, will be a topic for a future study.

Our second example is an omega band event identified by Partamies et al. (2017). Figure 21 shows an overview of the event with the peak times of three omegas at 01:31:00 UT, 01:36:00 UT, and 01:44:00 UT indicated by red vertical lines. The mapped ASC images and equivalent current patterns at these times are shown in Figure 22. There was a strong background WEJ to which the variations associated with the omegas were superposed. The PCA results of this 22 min time interval are shown in Figure 22. EOF1 and EOF3 describe the eastward propagating vortices of alternating polarity expected in an omega

band event (Opgenoorth et al., 1983). The tilted vortex pairs in EOF2 and EOF4 are less straightforward to interpret, but they could be associated with the interaction of the background current system with the disturbance created by the omega bands. Nonetheless, the four most significant EOFs in this omega band example are not the same as those we have obtained for substorm onsets.

## 5   Conclusions

We have examined the ionospheric equivalent currents and auroras during a subset of 28 substorm onsets identified by Frey et al. (2004) that occurred in the region best covered by the IMAGE magnetometers. We used the 2D SECS method to remove the internal contribution from the measured magnetic field and to calculate the ionospheric equivalent currents. We used PCA to characterize the spatiotemporal development of the ionospheric currents $\pm 10$ min around substorm onsets. Our main findings are as follows:

1. Ionospheric equivalent currents around substorm onsets can typically be described by three basic components: a westward electrojet (WEJ), a channel of poleward equivalent current, and a vortex. In addition, there may be a background current system (EEJ, WEJ, or Harang) associated with the large-scale convection.

2. A clockwise vortex (proxy for downward FAC) combined with the WEJ describes a WEJ with a localized equatorward bend, which typically appears before the substorm onset and is associated with an auroral arc. An anticlockwise vortex

(proxy for upward FAC) combined with the WEJ produces a WEJ with a localized poleward bend, which typically appears at the time of the substorm onset and is associated with a winding of the auroral arc into an auroral spiral. This is followed by decaying oscillations, where the polarity of the vortex changes. After this, a clockwise polarity of the vortex often occurs in combination of the WEJ, and is associated with an equatorward bend in the auroras.

3. The equivalent current channel (wedgelet) has been associated with BBFs by previous studies. The channel typically

starts to intensify before substorm onset and keeps intensifying during the examined 10 min after the substorm onset. The vortex and WEJ are located in the middle of the equatorward end of the channel.

4. Based on an analysis of a few other event types, it seems that dynamics of the WEJ, channel, and vortex are typical for substorm onsets.



5. Rapid geomagnetic variations at the substorm onset location, which can drive GIC in technological conductor networks, are mainly associated with the oscillating WEJ. The dynamics of the WEJ, vortex, and channel can describe up to 95% of the variance of the time derivative of the equivalent currents during the 20 min interval, indicating a possibility to predict substorm onset-related GIC.

6. A possible interpretation of the results is that a BBF from an X-line impacts the transition region between dipolar and tail-like field lines. The intensification and equatorward bend of the WEJ could be caused by the compression of the transition region, the subsequent poleward expansion by the rebound, and the decaying oscillations by oscillations of the transition region around its new balance point. The intensifying flow could further compress the transition region, producing the subsequent equatorward drift of the WEJ.

*Code and data availability.* IMAGE data are available at https://space.fmi.fi/image/www/?page=user_defined. The code for the SECS analysis is available in Vanhamäki and Juusola (2020). The code used to calculate magnetic coordinates and local times is available at https://apexpy.readthedocs.io/en/latest/ (Laundal et al., 2022). ASC data are available on request from Kirsti Kauristie (kirsti.kauristie@fmi.fi). The substorm onset list is available as auxiliary material to Frey et al. (2004).

*Author contributions.* LJ prepared the material and wrote the manuscript. AV provided expertise on analysis methods and geomagnetic induction, NP on ASC data, HV on ionospheric electrodynamics and substorm physics, MK on on the behavior of internal and external magnetic field variations, and SW on auroral electrojets. All co-authors read the manuscript and commented on it.

*Competing interests.* The authors declare that they have no conflict of interest.

*Acknowledgements.* We thank the institutes who maintain the IMAGE Magnetometer Array: Tromsø Geophysical Observatory of UiT the Arctic University of Norway (Norway), Finnish Meteorological Institute (Finland), Institute of Geophysics Polish Academy of Sciences (Poland), GFZ German Research Centre for Geosciences (Germany), Geological Survey of Sweden (Sweden), Swedish Institute of Space Physics (Sweden), Sodankylä Geophysical Observatory of the University of Oulu (Finland), Polar Geophysical Institute (Russia), DTU Technical University of Denmark (Denmark), and Science Institute of the University of Iceland (Iceland). We acknowledge the substorm timing list identified by Frey et al. (2004). This data is publicly available as auxiliary material to the named publication. This research was supported by the Academy of Finland project no. 339329 and the International Space Science Institute (ISSI) in Bern, through ISSI International Team project 506 (Understanding Mesoscale Ionospheric Electrodynamics Using Regional Data Assimilation).



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









**Figure 3.** External (ionospheric) equivalent current density ($\boldsymbol{J}$, arrows), its curl (interpreted as the vertical current density $j_z$, positive downward, color), and auroral intensity (greys) a few minutes around a substorm onset on 18 Dec 2001 at 21:59:00 UT. The ASC images have been mapped to 115 km altitude. Note that the scaling of the arrows and color background varies from panel to panel while the grey scale remains the same (black – 0 counts, white – 255 counts). The yellow star indicates the substorm onset location according to Frey et al. (2004). The black vertical line passing through the nearest station (AND) indicates the locations from which latitude profiles are extracted to create the time series representations in Fig. 2.





**Figure 4.** Four most significant empirical orthogonal functions (EOF1–EOF4) or spatial modes of $d\mathbf{J}/dt$ and $dj_z/dt$, their amplitudes or temporal modes, and time integrals of the amplitudes for $\pm 10\,\mathrm{min}$ around a substorm onset on 18 Dec 2001 at 21:59:00 UT. The cumulative percentage of the variance explained by each EOF is indicated in the parenthesis. The grey curves (117 curves) correspond to the rest of the EOFs having a smaller contribution.

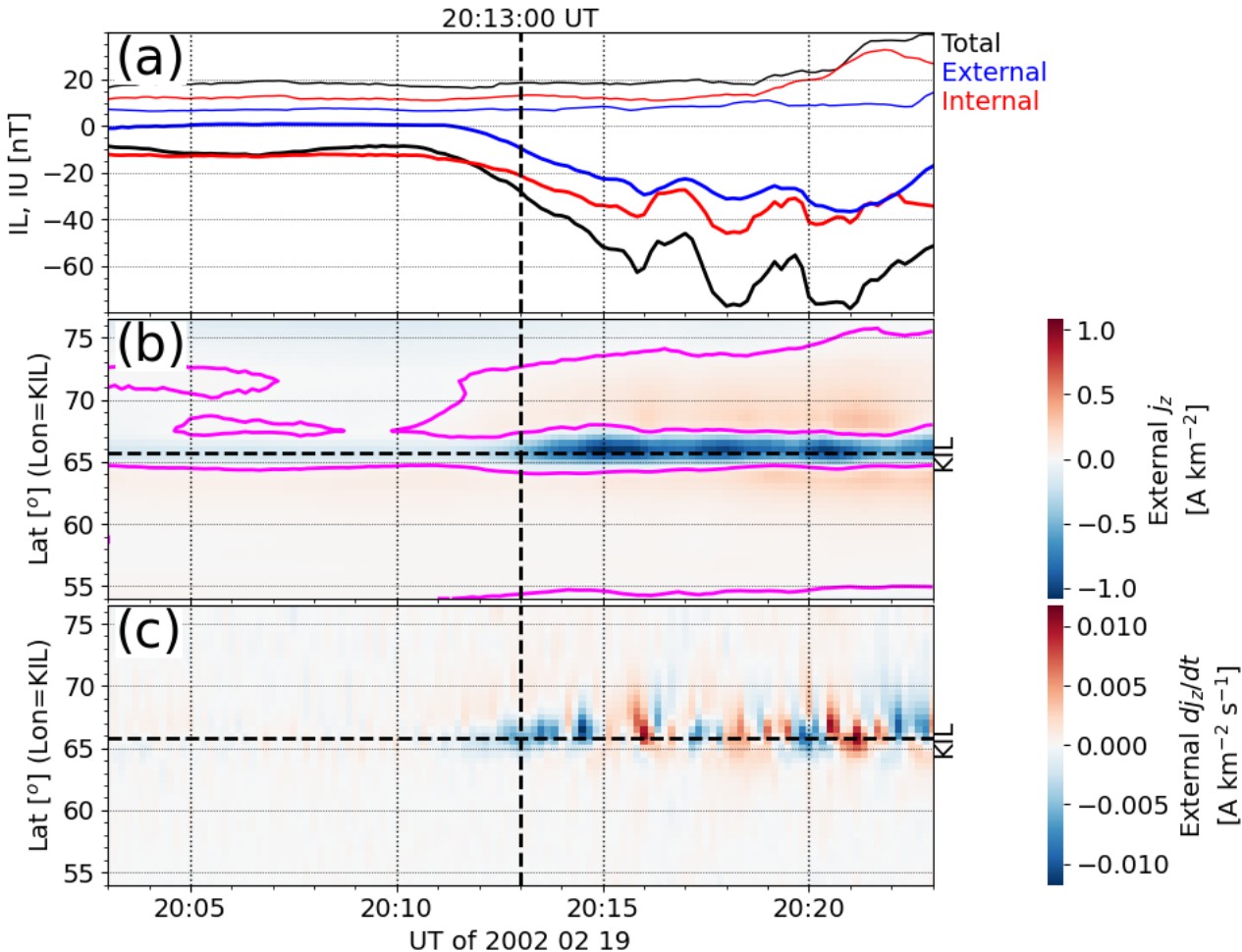

**Figure 5.** The same as Fig. 2, except for the substorm onset on 19 Feb 2002 at 20:13:00 UT.





**Figure 6.** The same as Fig. 3, except for the substorm onset on 19 Feb 2002 at 20:13:00 UT. The ASC images have been mapped to 120 km altitude.





**Figure 7.** The same as Fig. 4, except for the substorm onset on 19 Feb 2002 at 20:13:00 UT.





**Figure 8.** The same as Fig. 2, except for the substorm onset on 11 Mar 2002 at 21:06:00 UT.



**Figure 9.** The same as Fig. 3, except for the substorm onset on 11 Mar 2002 at 21:06:00 UT. The ASC images have been mapped to 120 km altitude.





**Figure 10.** The same as Fig. 4, except for the substorm onset on 11 Mar 2002 at 21:06:00 UT.



**Figure 11.** The same as Fig. 2, except for the substorm onset on 24 Jul 2001 at 20:33:00 UT.



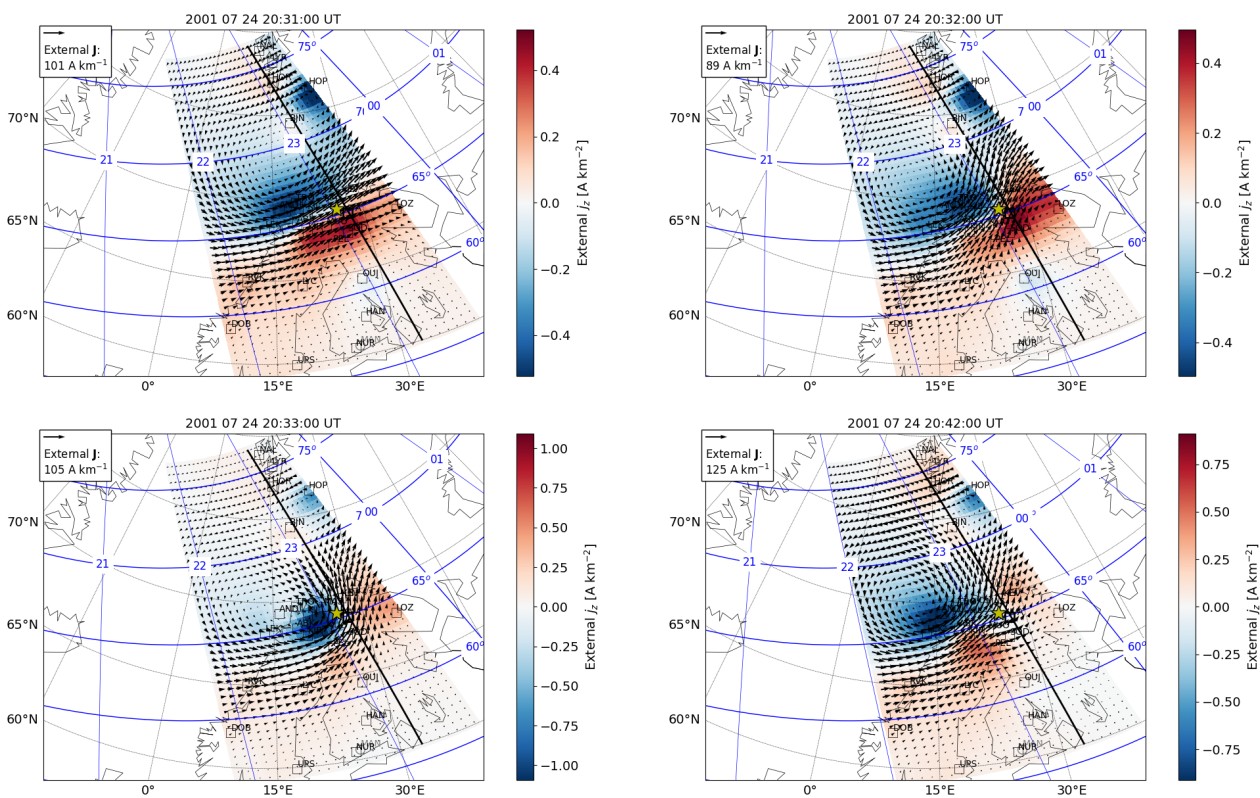

**Figure 12.** The same as Fig. 3, except for the substorm onset on 24 Jul 2001 at 20:33:00 UT and without ASC data.





**Figure 13.** The same as Fig. 4, except for the substorm onset on 24 Jul 2001 at 20:33:00 UT.





**Figure 14.** The same as Fig. 2, except for the substorm onset on 27 Jul 2002 at 01:23:00 UT.



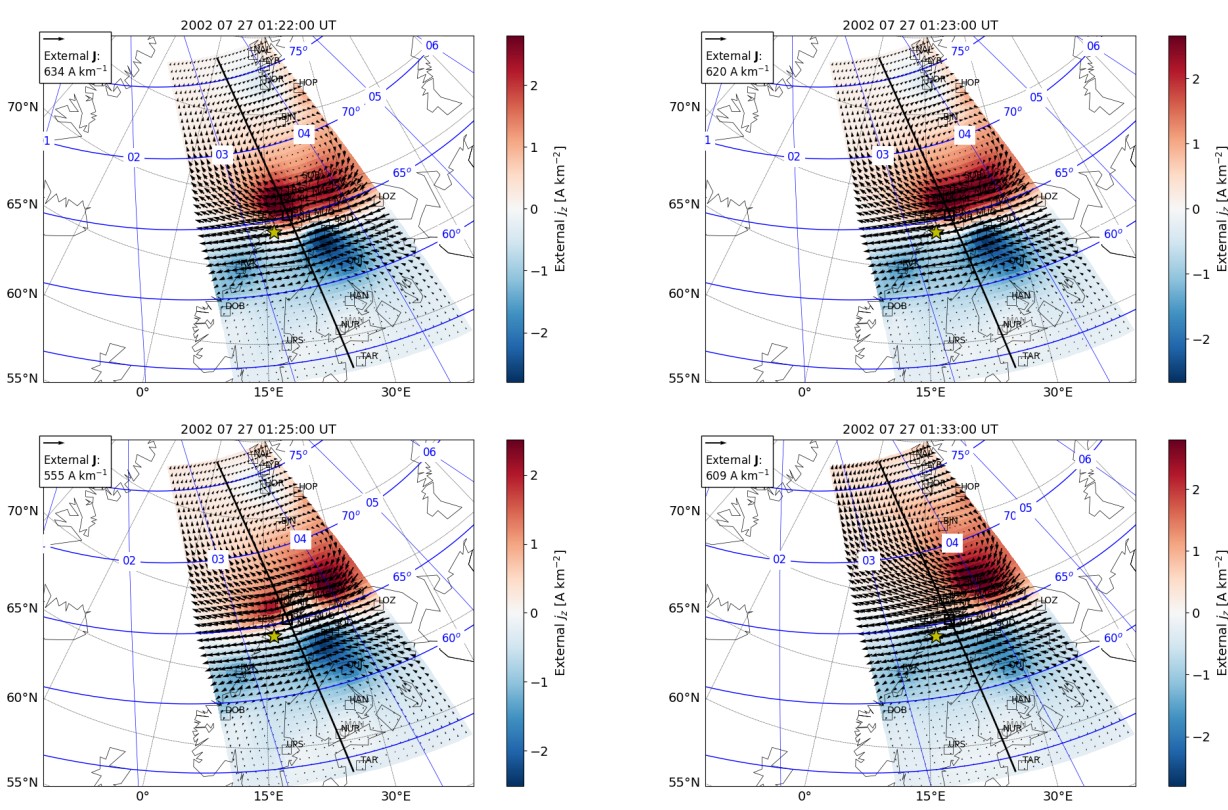

**Figure 15.** The same as Fig. 3, except for the substorm onset on 27 Jul 2002 at 01:23:00 UT and without the ASC data.





**Figure 16.** The same as Fig. 4, except for the substorm onset on 27 Jul 2002 at 01:23:00 UT.



**Table 2.** Substorm onsets according to Frey et al. (2004) that took place over the densest coverage of IMAGE. The columns are: number, onset time, onset location in geographic latitude and longitude, onset location in magnetic latitude and local time, nearest IMAGE station, background equivalent current system (EEJ – Eastward Electrojet, HD – Harang Discontinuity, WEJ – Westward Electrojet), description of the most significant Empirical Orthogonal Functions (EOFs) from the Principal Component Analysis (PCA), and percentage of the variance of $|d\mathbf{J}/dt|$ explained by four of the most significant EOFs. The events shown as examples in section 3 are indicated with an asterisk (*) in the first column.

| Number | Date HH:MM [UT] | (GLAT,GLON) ([deg],[deg]) | (MLAT,MLT) ([deg],[h]) | Station | Background | PCA EOFs | Explained [%] |
|---|---|---|---|---|---|---|---|
| 1. | 2002 Oct 09 15:59 | (70.48,24.74) | (66.89,18.65) | SOR | EEJ/HD | WEJ/EEJ, channel, vortex | 88 |
| 2. | 2000 Sep 17 19:10 | (69.05,19.22) | (65.77,21.46) | KIL | HD/EEJ | WEJ, channel, vortex | 80 |
| 3. | 2001 May 15 18:54 | (70.11,26.95) | (66.41,21.79) | IVA | EEJ/HD | WEJ, channel, vortex | 82 |
| 4. | 2002 May 22 19:12 | (70.38,22.36) | (66.93,21.89) | SOR | EEJ/HD | WEJ, channel, vortex | 77 |
| *5. | 2002 Feb 19 20:13 | (69.26,23.81) | (65.72,22.28) | KIL | HD/EEJ | WEJ, channel, vortex | 88 |
| 6. | 2000 Oct 23 20:07 | (70.02,19.68) | (66.72,22.48) | TRO | Quiet | ? | 83 |
| 7. | 2000 Jul 13 19:48 | (68.96,26.52) | (65.28,22.50) | MAS | HD/EEJ | ? | 82 |
| 8. | 2002 Aug 23 20:48 | (69.49,18.53) | (66.26,23.02) | TRO | Quiet | WEJ, channel, vortex | 88 |
| 9. | 2001 Sep 13 20:34 | (68.05,24.80) | (64.44,23.08) | MUO | HD/WEJ | WEJ, channel, vortex | 87 |
| *10. | 2001 Jul 24 20:33 | (68.90,25.58) | (65.25,23.16) | IVA | EEJ | WEJ, channel, vortex | 84 |
| *11. | 2002 Mar 11 21:06 | (68.70,23.13) | (65.19,23.26) | MAS | Quiet | WEJ, channel, vortex | 93 |
| 12. | 2000 Dec 23 21:41 | (67.79,16.07) | (64.68,23.26) | ABK | EEJ/HD | WEJ, channel, vortex | 87 |
| 13. | 2001 Jan 20 21:40 | (68.43,18.23) | (65.20,23.27) | ABK | WEJ | WEJ, channel, vortex | 94[1] |
| 14. | 2001 Jul 15 21:03 | (67.97,19.71) | (64.64,23.33) | KIR | Quiet | ? | 85 |
| 15. | 2002 May 10 21:11 | (67.27,22.40) | (63.77,23.69) | PEL | WEJ | WEJ, channel, vortex | 83 |
| 16. | 2002 May 06 20:59 | (67.87,26.26) | (64.18,23.70) | SOD | WEJ | WEJ, channel, vortex | 85 |
| *17. | 2001 Dec 18 21:59 | (69.48,16.74) | (66.37,23.73) | AND | Quiet | WEJ, channel, vortex | 95 |
| 18. | 2000 Nov 05 21:19 | (68.91,26.23) | (65.24,23.88) | KEV | WEJ | WEJ, channel, vortex | 80 |
| 19. | 2002 Sep 08 21:26 | (68.78,26.80) | (65.07,0.06) | IVA | ? | ? | 82 |
| 20. | 2001 May 12 21:20 | (67.71,26.39) | (64.02,0.08) | SOD | WEJ | EEJ/WEJ, channel, vortex | 85 |
| 21. | 2000 Aug 07 22:11 | (68.07,19.38) | (64.75,0.36) | ABK | WEJ | WEJ, channel, vortex | 95 |
| 22. | 2001 May 18 21:56 | (69.60,21.75) | (66.18,0.53) | KIL | Quiet | WEJ, channel, vortex | 88 |
| 23. | 2002 Jul 16 22:25 | (67.47,21.77) | (64.00,0.75) | MUO | WEJ | WEJ, channel, vortex | 86 |
| 24. | 2001 Aug 13 22:34 | (67.57,24.23) | (63.98,0.96) | MUO | WEJ | WEJ, channel, vortex | 89 |
| 25. | 2001 Jun 30 23:36 | (68.64,17.19) | (65.47,1.78) | ABK | WEJ | WEJ, channel, vortex | 91 |
| 26. | 2002 Mar 21 23:35 | (70.38,25.44) | (66.76,2.00) | KEV | HD/EEJ | ? | 88 |
| 27. | 2002 May 21 00:03 | (68.34,17.51) | (65.15,2.32) | ABK | WEJ | WEJ, channel, vortex | 95[1] |
| *28. | 2002 Jul 27 01:23 | (67.32,16.35) | (64.17,3.29) | ABK | WEJ | WEJ, channel, vortex | 80 |

[1] Five of the most significant EOFs included instead of four.



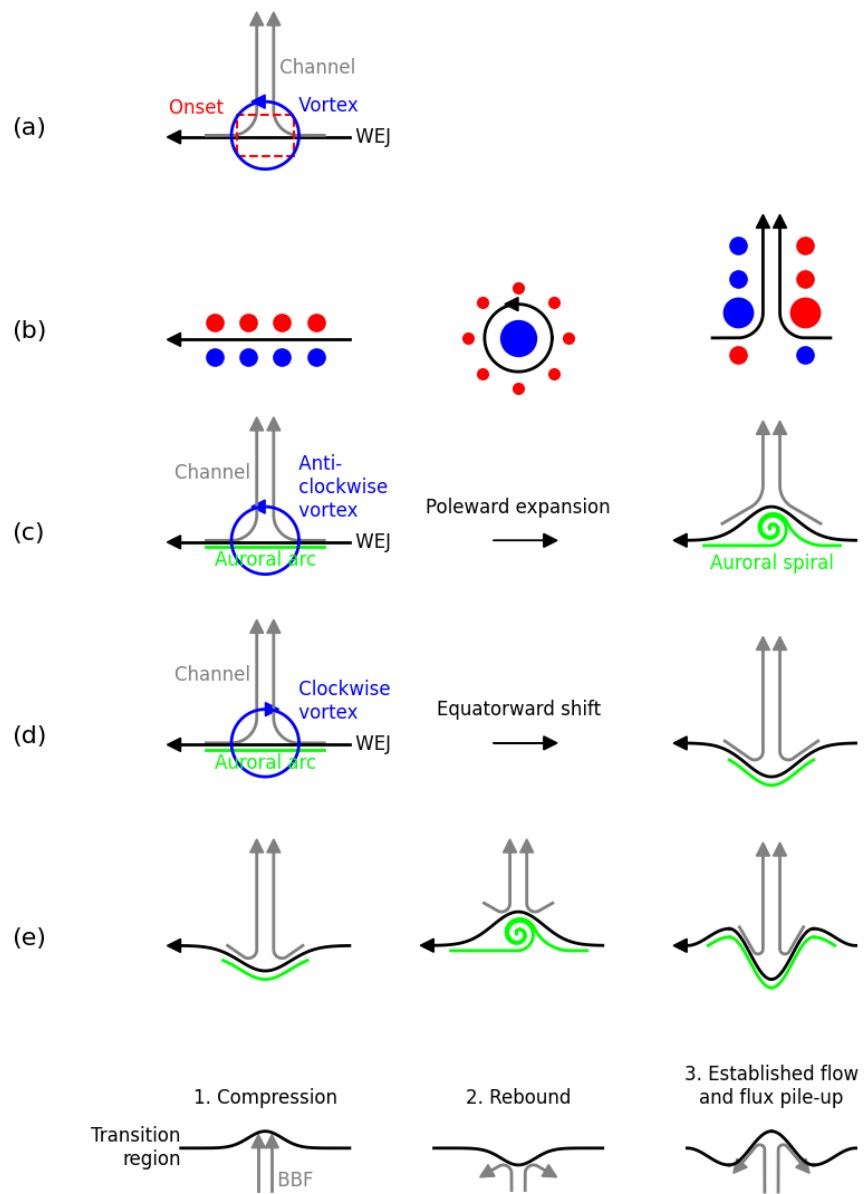

**Figure 17.** (a): Three basic components of ionospheric equivalent currents observed around substorm onsets: a WEJ, a channel of poleward current (with occasionally a weaker channel of equatorward current equatorward of it), and a vortex. (b): Curl of the equivalent current density associated with the WEJ, vortex, and channel. Negative curl can be interpreted as upward FAC and positive curl as downward FAC under conditions outlined in section 2.4 (c): Superposition of a WEJ and and an anticlockwise vortex will produce a poleward bulge in the WEJ. The auroral arc associated with the WEJ is wound into a spiral by the strong upward FAC in the center of the vortex. (d): Superposition of a WEJ and a clockwise vortex will produce an equatorward indentation in the WEJ. The auroral arc associated with the WEJ will be also be shifted equtorward by the strong downward FAC in the center of the vortex. (e): A possible interpretation of the results in terms of magnetospheric processes.





**Figure 18.** Total, external, and internal geomagnetic north ($B_x$), east ($B_y$), and down ($B_z$) components and their time derivatives ($dB_x/dt$, $dB_y/dt$, and $dB_z/dt$) at station AND ±10 min around a substorm onset on 18 Dec 2001 at 21:59:00 UT.







**Figure 19.** The same as Fig. 2, except for a longer time interval from −30 min to +120 min around the substorm onset.







**Figure 20.** The same as Fig. 4, except for a longer time interval from $-30\,\mathrm{min}$ to $+120\,\mathrm{min}$ around the substorm onset.

**Figure 21.** The same as Fig. 2, except for an omega band event on 29 Mar 2001. The peak times of three omegas (01:31:00 UT, 01:36:00 UT, and 01:44:00 UT) are indicated with the red vertical lines.



**Figure 22.** The same as Fig. 3, except for an omega band event on 20 Mar 2001. The ASC images have been mapped to 110 km altitude.



**Figure 23.** The same as Fig. 4, except for an omega band event on 29 Mar 2001. The peak times of three omegas are indicated with the red vertical lines.