# Peer review of "Three principal components describe the spatiotemporal development of meso-scale ionospheric equivalent currents around substorm onsets"

_EGUsphere, 2023_

## Author Response (AR1)

**Replies to Referees**

Liisa Juusola[1], Ari Viljanen[1], Noora Partamies[2], Heikki Vanhamäki[3], Mirjam Kellinsalmi[1], and Simon Walker[4]

[1]Finnish Meteorological Institute, Helsinki, Finland
[2]Department of Arctic Geophysics, University Centre in Svalbard (UNIS), Longyearbyen, Norway
[3]University of Oulu, Oulu, Finland
[4]University of Bergen, Bergen, Norway

**Referee 1**

**Dear Professor Lyons,**

thank you for very useful and constructive comments. Please see below for our point-by-point replies. The original review is written in *black* and our replies in blue. The line numbers refer to the revised version of the manuscript with the changes marked.

*This paper first describes the ionospheric equivalent current density associated with substorm onsets as identified using spacecraft global auroral images by Frey et al. [2004]. They then use Principal Component Analysis to characterize the current system changes associated with the onset. This analysis is done on the time derivative of the equivalent currents. If I understand correctly, this means that an EOF that oscillates, as in seen in some cases in the paper, implies a current system that is varying in strength, but not one that is oscillating (I think something about this should be mentioned in the paper). The description of the analysis procedure, and its limitations, is very clear, and includes several useful references. As described in the text, they found that the equivalent currents can typically be described by three components: a channel of poleward equivalent current (wedgelet), a westward electrojet (WEJ) associated with an auroral arc, and a vortex. The WEJ and vortex are located at the equatorward end of the channel, which has been associated with tail flow bursts and auroral streamers by previous studies. The results are interesting and certainly warrant publication.*

*It is impressive to me that evidence for the flow channel leading to onset shows so clearly in the equivalent currents. Perhaps this could be given more emphasis. In addition, I would like to encourage the authors to also consider another aspect of their observations that is directly related to the evolution of flow channels in the inner plasma sheet and how the flow channel leads to onset, which is east-west aligned. Rather than locally pushing an arc equatorward as illustrated in Figure 17d, I believe that the paper's results support the two-dimensional picture initially shown in the schematic Figure 14 put together from radar and ASI observation in Zou et al. (2009). The Rice Convection Model shows that, as a result of the energy-dependent magnetic drift, the low entropy plasma of a flow burst spreads azimuthally within the inner plasma (Wang et al., 2018; Yang et al., 2014), which would lead to the Zou et al. picture and the east-west oriented onset. Our group has more recently given evidence for this flow-channel spreading (Lyons et al., 2022, 2021) using ground-based radar and all sky imager observations, which show subauroral polarization stream (SAPS) and dawnside auroral polarization stream (DAPS) (see Liu et al., 2020) enhancements at appropriate locations relative to onset. This pattern has the incoming flow channel flows diverting to the dawnside Region 1 downward current region (poleward of most dawnward auroral activity) and to the duskside Region 2 downward current region (equatorward of most duskward auroral activity). This can be seen in the 2159:10 UT panel of Figure 3, the combination of EOF2 and EOF4 in Figure 4, the 2013:10 panel of Figure 6, the combination of EOF3 and EOF4 in Figure 7, and the 2106:10, 2106:50, 2107:50 panels of Figure 9. The final 2 events in the paper do not have significant ground magnetic field changes associated with the auroral brightening, so we cannot see anything for those events. I believe the above substantially affects the*

*speculative discussion near the end of the paper and the schematic Figure 17 and instead supports the 2-D schematic Figure 14 in Zou et al. (2009).*

We have studied the suggested references and agree that they are very relevant. Thank you. Especially, the Rice Convection model can explain how a BBF approaching the transition region creates the auroral arc. We agree that the equivalent current channel pattern resembles that expected for the flows, including the SAPS and DAPS signatures. This has led to several changes throughout the text: lines 51–66, 257–268, 274–275, 283–284, 291, 304, 328–330, 374–375, and Figure 17.

Although the Rice Convection Model description of a BBF approaching the transition region can explain the equivalent current channel, auroral arc, possibly the SAPS and DAPS signatures in the equivalent current, and the westward motion of the channel (or its westward edge), it does not seem to explain the spiral or WEJ. Thus, we have modified the manuscript in the following way: We have applied the RCM model of a BBF interacting with the transition region, including discussion on SAPS and DAPS, to explain the equivalent current channel pattern, its development, and the auroral arc. The schematic picture in Figure 17 does not disagree with this interpretation, because it merely attempts to explain the observed changes in the latitudinal location of the signatures (more equatorward in the ionosphere indicating more earthward in the magnetosphere and more poleward in the ionosphere indicating more tailward in the magnetosphere). However, we have modified the figure to include the SAPS and DAPS signatures.

*The discussion on the EOFs in lines 237–245 could be made more precise. EOF 2 describes a north-south channel that connects dawnward as an azimuthal channel in the downward current region, and EOF4 connects both dawnward as an azimuthal channel and duskward as an azimuthal channel and lower latitudes. Also, here might be a good place to point out the EOF2 oscillations are not in the direction of the current but just in the time rate of change of total current magnitude.*

This is a good suggestion, and we have modified the text as suggested (lines 257–268).

*Why is the direction of the vortex current of EOF1 in Figure 7 and the associated field-aligned current direction opposite to those of EOF3 in Figure 4? In both the examples in Figure 5 and Figure 8, the downward and upward field-aligned currents enhance at onset, as for the first example (Figure 2).*

The signs of an EOF and its amplitude are interchangeable. The relevant quantity is their product. EOF1 in Figure 7 has a negative amplitude at substorm onset, which results in a field-aligned current in the same direction as that of EOF3 in Figure 4 (lines 291–293).

*To me, the motivation is not clear for including the final two examples that do not show substantial magnetic field changes associated with the onset.*

Our motivation for this was to show examples during all different background conditions and to demonstrate that even such cases produce the typical magnetic field variations even though they are masked by the strong background current system. We have tried to explain this at the beginning of section 3.

**Referee 2**

**Dear Reviewer,**

thank you for a positive review of our manuscript. Please see below for our reply to the question you posed. The original review is written in *black* and our replies in blue. The line numbers refer to the revised version of the manuscript with the changes marked.

*Review of : "Three principal components describe the spatiotemporal development of meso-scale ionospheric equivalent currents around substorm onsets" by Juusola et al.*

*This work deals with a very complex process, and topical issue that is substorms mechanism, which is a controversial topic in space physics. It presents a thorough analysis of several cases and interpretation of data analysis using PCA. I consider it will be a valuable contribution to the field of magnetosphere-ionosphere interactions adding to the understanding of substorm physics, and also in the field of Space Weather.*

*I have just one comment.*

*I understand that depending on the example analyzed, the EOF1 to 4 represents different current patterns. For example, in Example 1 EOF1 describes a WEJ and in Example 2 this corresponds to EOF2. In Example 3, you mention that EOF1 may*

*describe the vortex and WEJ. And from what you mention in Section 4.4 regarding substorm onset identification, what happens if I consider for the PCA analysis the same period length you use in your analysis (20 minutes), but beginning 20 minutes prior to the substorm occurrence, that us your series will end at the exact time the substorm is beginning or close to it. Will you*
80  *obtain EOFs, or some of them, that you obtain now that could indicate that substorm is going to occur? I mean, will a WEJ or a vortex appear in the statistical analyisis, if the time series I use have not yet evolved as a substorm?*

If you analyse a 20 min time series that ends at substorm onset, for example 21:39-21:59 UT on 18 December 2001 (Example 1), you will get the WEJ and the channel among the four most significant EOFs, but not the spiral. This is because, as you can see from Fig. 4, those are the spatial features present in the time series. We tested this, and the time development of the WEJ
85  and channel amplitudes was similar to the corresponding EOFs in Fig. 4 before the onset.

**Community comments**

**by Dr. James Weygand**

We are grateful to Dr. Weygand for reading the manuscript and for providing very good suggestions on how to improve it. Please see below for our point-by-point replies to the comments. The original review is written in *black* and our replies in blue.
90  The line numbers refer to the revised version of the manuscript with the changes marked.

*Line 48: I feel a numerical value should be given here to provide the reader with a basis of comparison later during the review of the examples.*

Unfortunately we could not find general numerical values in the literature. Most likely the times vary from event to event.

*Line 70: In my personnel communication with Harald Frey he has stated that there are most likely pseudo breakups and*
95  *poleward boundary intensifications within his list. I believe the Weygand et al. [2008] study alludes to this potential problem. I feel it is worth noting but do not believe it will change your results.*

Thank you, we have added this at lines 96–97.

*Line 225: "At 21:59:40 UT, the arc had developed a poleward bulge" Is this part of the westward traveling surge? If so, it would be worth identifying this for the reader to help them interpret these current maps. If the westward surge is observed*
100  *elsewhere then please identify it for the reader.*

Most likely it is, we have added this at lines 242–243.

*This is just a comment and not a request to include or for a change, but the real event shown in figure 3 looks very similar to the modelled event shown in Baumjohann and Glaßmeier (1984).*

Thank you, we have added this at lines 248–249
105  *Line 230-231: There is a comment about the hall currents being anti-parallel to the ionospheric flow. Are SuperDARN measurements available for these events to support the derivation of the current maps over region where there is little to know magnetometer coverage? If so, then do the ionospheric flow observations generally support the equivalent ionospheric currents. I understand that the equivalent current are not the Hall currents, but they should be close.*

As far as we understand it, there are no merged velocity vectors available for the IMAGE area for our example events.
110  *Line 312-313: It is difficult here and in some of the other cases why EOF1 is identified as a vortex and EOF2 is a westward electrojet. Is that because the equivalent currents in EOF2 are large magnitudes all the way from the east to the west where as in EOF1 they are large in the "vortex" but weaker in the west edge of the map. It looks like one could say there is a westward electrojet in both.*

Yes, that is how they have been identified, and we have added this explanation at lines 344–348. Thank you. It is also true
115  that in some cases the identification is somewhat ambiguos. This is discussed in section 4.1, where we conclude that although the spiral, WEJ, and channel may get entangled in the EOFs, their existense can still be distinguished in the majority of the events.

*Line 355-357: I feel the reader would greatly benefit from having a movie of this event to better see the time development and motion of the EOF currents.*

120      This would be a nice addition, but unfortunately we do not have the resources to make one. We have tried to illustrate the development with Fig. 17. We have added references to appropriate panels in the figure to make the text more clear (lines 388–402).

**References**

Baumjohann, W. and Glaßmeier, K.-H.: The transient response mechanism and Pi2 pulsations at substorm onset—Review and outlook, Planetary and Space Science, 32, 1361–1370, https://doi.org/https://doi.org/10.1016/0032-0633(84)90079-5, 1984.

Liu, J., Lyons, L. R., Wang, C.-P., Hairston, M. R., Zhang, Y., and Zou, Y.: Dawnside auroral polarization streams, Journal of Geophysical Research: Space Physics, 125, e2019JA027742, https://doi.org/https://doi.org/10.1029/2019JA027742, 2020.

Lyons, L. R., Nishimura, Y., Wang, C.-P., Liu, J., and Bristow, W. A.: Two-dimensional structure of flow channels and associated upward field-aligned currents: model and observations, Frontiers in Astronomy and Space Sciences, 8, https://doi.org/https://doi.org/10.3389/fspas.2021.737946, 2021.

Lyons, L. R., Nishimura, Y., Liu, J., Bristow, W. A., Zou, Y., and Donovan, E. F.: Verification of substorm onset from intruding flow channels with high-resolution SuperDARN radar flow maps, Journal of Geophysical Research: Space Physics, 127, e2022JA030723, https://doi.org/https://doi.org/10.1029/2022JA030723, 2022.

Wang, C.-P., Gkioulidou, M., Lyons, L. R., and Wolf, R. A.: Spatial distribution of plasma sheet entropy reduction caused by a plasma bubble: Rice Convection Model simulations, Journal of Geophysical Research: Space Physics, 123, 3380–3397, https://doi.org/https://doi.org/10.1029/2018JA025347, 2018.

Yang, J., Toffoletto, F. R., and Wolf, R. A.: RCM-E simulation of a thin arc preceded by a north-south-aligned auroral streamer, Geophys. Res. Lett., 41, 2695–2701, https://doi.org/https://doi.org/10.1002/2014GL059840, 2014.

Zou, S., Lyons, L. R., Nicolls, M. J., Heinselman, C. J., and Mende, S. B.: Nightside ionospheric electrodynamics associated with substorms: PFISR and THEMIS ASI observations, J. Geophys. Res., 114, https://doi.org/https://doi.org/10.1029/2009JA014259, 2009.

---

## Author Response (AR2)

**Replies to Referees**

Liisa Juusola[1], Ari Viljanen[1], Noora Partamies[2], Heikki Vanhamäki[3], Mirjam Kellinsalmi[1], and Simon Walker[4]

[1]Finnish Meteorological Institute, Helsinki, Finland
[2]Department of Arctic Geophysics, University Centre in Svalbard (UNIS), Longyearbyen, Norway
[3]University of Oulu, Oulu, Finland
[4]University of Bergen, Bergen, Norway

**Referee 1**

Professor Lyons wrote: *Thank you for giving thoughtful consideration of my first review. I have no further comments other than I noticed a typo in line 243. I believe that the time should be 22:00:40, no 22:40:00.*

  Thank you for noticing this, we have corrected the time.